# Evaluation of Low-Cost Multi-Spectral Sensors for Measuring Chlorophyll Levels Across Diverse Leaf Types

**DOI:** 10.3390/s25072198

**Published:** 2025-03-31

**Authors:** Prattana Lopin, Pichapob Nawsang, Srisangwan Laywisadkul, Kyle V. Lopin

**Affiliations:** 1Department of Biology, Faculty of Science, Naresuan University, Phitsanulok 65000, Thailand; prattanas@nu.ac.th (P.L.); srisangwanl@nu.ac.th (S.L.); 2Center of Excellence for Biodiversity, Faculty of Science, Naresuan University, Phitsanulok 65000, Thailand; 3Department of Physics, Faculty of Science, Naresuan University, Phitsanulok 65000, Thailand

**Keywords:** chlorophyll sensors, low-cost sensors, spectral sensors, spectral reflectance, partial least squares regression, nested cross-validation, learning curves, model validation, machine learning

## Abstract

Chlorophyll levels are a key indicator of plant nitrogen status, which plays a critical role in optimizing agricultural yields. This study evaluated the performance of three low-cost multi-spectral sensors, AS7262, AS7263, and AS7265x, for non-destructive chlorophyll measurement. Measurements were taken from a diverse set of five leaf types, including smooth, uniform leaves (banana and mango), textured leaves (jasmine and sugarcane), and narrow leaves (rice). Partial least squares regression models were used to fit sensor spectra to chlorophyll levels, using nested cross-validation to ensure robust model evaluation. Sensor performance was assessed using $R^{2}$ and mean absolute error (MAE) scores. The AS7265x demonstrated the best performance on smooth, uniform leaves with validation $R^{2}$ scores of 0.96–0.95. Its performance decreased for the other leaves, with $R^{2}$ scores of 0.75–0.85. The AS7262 and AS7263 sensors, while slightly less accurate, achieved reasonable $R^{2}$ scores ranging from 0.93 to 0.86 for smooth leaves, and from 0.85 to 0.73 for the other leaves. All sensors, particularly the AS7265x, show potential for non-destructive chlorophyll measurement in agricultural applications. Their low cost and reasonable accuracy make them suitable for agricultural applications such as monitoring plant nitrogen levels.

## 1. Introduction

Agricultural output increased dramatically in the mid-20th century due to advancements in crop breeding, irrigation, pesticides, and fertilizers. The development and usage of nitrogen, phosphorus, and potassium fertilizers allowed farmers to supply nutrients required by high-yield crops. Nitrogen is a critical macronutrient for plant growth, as it is concentrated in the photosynthesis components of leaves. In nitrogen-deficient soils, plants can allocate up to 40–50% of their energy to the roots and below-ground biomass [1], with a substantial portion of that energy dedicated to biological nitrogen fixation or the active uptake of mineralized nitrogen [2].

While only 2–5% of leaf nitrogen is found in chlorophyll, chlorophyll content serves as an indicator of plant nitrogen levels through a feedback mechanism. As nitrogen availability decreases, the major nitrogen-containing components of photosynthesis, such as RuBisco and thylakoid proteins, become limited. Consequently, plants reduce chlorophyll levels because they cannot process the excess captured light in a low-nitrogen state. This makes chlorophyll an indicator of plant nitrogen status. This relationship has led to numerous studies exploring the use of chlorophyll meters to monitor whole-plant nitrogen levels [3].

A variety of methods are used to measure plant chlorophyll levels. For ease of use, widespread application, and rapid measurement, the optical properties of the leaves are often utilized. These methods rely on chlorophyll’s characteristic absorption of light, with higher absorption in the blue and red spectrum compared to the green spectrum, or its absorption in the red region coupled with high reflectance in the near-infrared region (NIR) [4]. Chlorophyll estimate techniques range widely, from hyperspectral cameras on satellites [5] and unmanned aerial vehicles (UAVs) [6] to handheld multi-spectral cameras [7], handheld chlorophyll meters [8], and even simple printed color charts [9].

Commercially available handheld chlorophyll meters, with the SPAD (Soil Plant Analysis Development) meter (Konica Minolta, Tokyo, Japan) being the most popular and the atLEAF meter (FT Green LLC, Wilmington, DE, USA) also widely used [10], rely on measuring the transmission of light in the red and NIR regions to calculate a chlorophyll content index (CCI). Numerous studies have correlated these CCIs with chlorophyll content across multiple plant species [11]. These handheld devices measure a small section of a single leaf, providing localized data. Additionally, canopy sensors such as GreenSeeker and Crop Circle are available. These sensors measure spectral reflectance from a large area, allowing for the assessment of chlorophyll levels across entire fields [12].

As these sensors can estimate chlorophyll content, they can also serve as a proxy to assess plant nitrogen levels [13]. This capability has been applied in studies using chlorophyll meters to optimize nitrogen fertilizer application [14]. Proper nitrogen application is critical as insufficient nitrogen limits agricultural yields. However, excessive nitrogen increases production cost [15], contributes to pollution through nitrogen runoff [16], and raises greenhouse gas emissions due to the production of nitrous oxide [17]. Moreover, excess nitrogen can reduce yields by promoting excess vegetative growth at the expense of the plant’s agricultural produce [18]. This phenomenon has been seen in perennial and annual crops [19].

To balance the need to add nitrogen to soils to increase yields while avoiding excessive application, site-specific nitrogen management (SSNM) has been developed [20]. SSNM uses chlorophyll level measurements with algorithms to optimize nitrogen application rates [21,22]. Studies on SSNM have been conducted on a wide range of crops, ranging from pear orchards [23] to rice paddies [24]. Given the significant variation in crop types, locations, and climates studied, SSNM results cannot be easily generalized. However, specific SSNM examples include a 40–45% reduction in nitrogen input in wheat production [25] and an 18% reduction in nitrogen use for rice while increasing yields by 8% and generating an 11% increase in economic returns [26]. Additional examples and insights into SSNM can be found in recent comprehensive reviews [27,28].

Despite significant advances in agricultural production, some challenges remain. One major issue is that small-scale farmers often cannot fully benefit from these advancements due to their cost or complexity. This is a substantial concern, as there are an estimated 450 million small-scale farms worldwide, comprising 40% of global agricultural land and supporting 2.2 billion people [27,29]. Due to limited access to capital and resources, small-scale farmers often experience lower yields compared to larger farms. To compensate for this, they frequently rely on increased use of variable inputs, such as nitrogen fertilizer, to boost yields [30]. For example, in China, every 1% increase in farm size results in a reduction in fertilizer use by 0.3% [31].

Research has focused on developing low-cost chlorophyll sensors to meet the needs of small-scale farms. Examples include the use of smartphones to estimate chlorophyll content [32]. However, most studies correlate smartphone results to SPAD values rather than directly to chlorophyll levels [33,34]. Similarly, many studies have investigated handheld chlorophyll meters that utilize a red LED and a NIR LED, similar to the SPAD meter, correlating the low-cost sensor output with SPAD measurements rather than direct chlorophyll levels [35,36]. Additional research has employed red, green, and blue color sensors, again relating their results to SPAD CCI values, not to actual chlorophyll concentrations [37].

AMS multi-spectral sensors are also emerging as valuable tools for making low-cost devices to measure chlorophyll levels. Recent reports have demonstrated their potential in various applications. One study used the 11-channel AS7341 color sensor to measure chlorophyll levels in Hami melon leaves, achieving a test $R^{2}$ score of 0.87 [38]. Another study combined the 6-channel AS7262 and AS7263 sensors to measure chlorophyll levels in wheat leaves ($R^{2}$ = 0.63) and soybean leaves ($R^{2}$ = 0.82) [39]. Additionally, the AMS 14-channel spectral sensor AS7343 was used to measure chlorophyll levels in basil, though it produced a lower $R^{2}$ score < 0.2. However, this sensor performed well in fitting phenolic and flavonoid content, achieving $R^{2}$ scores > 0.87 [40].

AMS sensors operate by utilizing integrated optical filters, allowing the sensors to remain low-cost while providing moderate spectral resolution. The AMS AS7262 measures six visible light bands (450–650 nm), while the AS7263 measures six red-NIR bands (650–870 nm). The AS7265x combines three 6-channel sensors, allowing it to measure 18 spectral bands across the visible and NIR regions (410–940 nm). These sensors are significantly more affordable than traditional chlorophyll meters, with the AS7262 and AS7263 breakout boards priced at USD 28 each and the AS7265x at USD 70 from Sparkfun (as of January 2025). In comparison, the SPAD meter costs over USD 1000, and the atLEAF meter is around USD 250.

The visible and NIR regions covered by AMS sensors make them versatile tools for a wide range of applications. Agricultural products are the most prominent focus with AMS sensors being used to assess the quality of grapes [41], olives [42], oranges [43], dates [44], tomatoes [45,46], and even fish [47]. Beyond agricultural applications, AMS sensors have been used to monitor the development of rock lobsters [48], photosynthetically active radiation (PAR) [49,50], ocean respiration [51], and aid in weed detection [52].

In this study, we aim to evaluate the performance of the AMS AS7262, AS7263, and AS7265x sensors in measuring leaf chlorophyll levels across a diverse set of plants: banana, jasmine, mango, rice, and sugarcane.

## 2. Materials and Methods

### 2.1. Device Design and Characterization

To estimate chlorophyll levels, spectral measurements were first collected from leaves using AS7262, AS7263, and AS7625x sensors from AMS (Premstätten, Austria). Since these sensors require a microcontroller for data acquisition and control, breakout boards with mounted sensors were obtained from SparkFun (Boulder, CO, USA). The boards also featured a Qwiic connection that supports I^2^C communication, making it easy to connect with any Qwiic-compatible microcontrollers, which are also available from SparkFun. In this study, the SparkFun RedBoard Artemis and Artemis Nano were used interchangeably. A Qwiic multiplexer (Mux) was used to connect all three sensors to a single Arduino. The Mux is necessary because the sensors share the same I^2^C address, preventing them from being connected to the same I^2^C bus simultaneously. The Mux allows the micro-controller to use each sensor individually.

A diagram of the setup is shown in Figure 1, and a photograph of the device is shown in Figure 2. Each sensor had a button attached to start reading the sensor. The buttons were also mounted on Qwiic-connected breakout boards. Qwiic cables were used to connect the buttons to the sensors, the sensors to the Mux, and the Mux to the Arduino, while the Arduino was connected to a computer with a USB-C cable. Short Qwiic cables are shown for aesthetics in Figure 2, though during measurements, 500 mm cables were used for ease of use.

Electrical measurements of the AS7265x LEDs showed that the breakout boards could not accurately drive high currents (50 and 100 mA) through the white LED or even low currents through the UV LED. See Appendix A.

A custom 3D-printed holder was created to house the Arduino and Mux. In addition, individual holders for each button–sensor pair were 3D-printed for ease of use and to make a shroud around the light path. The STL and SKP files for the 3D-printed sensor holders are available in the 3d_files folder of the GitHub repository at https://github.com/KyleLopin/asm_chloro_test/(accessed on 25 March 2025).

### 2.2. Device Firmware and Software

To control the sensors and acquire data, a custom Arduino program was developed. The code would poll through each sensor and check whether a button was pressed. When a button is pressed, that sensor starts measuring. Different integration times and currents were tested to characterize the device and find the best conditions. To simplify the process, these conditions were hard-coded into the firmware. A Python program (v3.12) was also created to make a graphical user interface (GUI) for visualizing and saving the data as they were recorded. The Arduino and GUI code can be found in the source folder of the GitHub repository.

### 2.3. Leaf Selection and Spectra

To evaluate the sensors’ performance across a diverse range of leaves, spectral measurements were taken from different species and developmental stages. A total of 100 leaves from five different species (500 leaves in total) were selected for these experiments. To ensure a broad range of varying leaf ages and chlorophyll levels, approximately one-third of the samples were new leaves, one-third were from the middle of the plant, and one-third were older leaves. Photographs of a newer (light) and older (darker) leaf for all five species tested are shown in Figure 3.

The AS7262 measures six spectral channels in the visible range (450, 500, 550, 570, 600, and 650 nm) with a Full Width at Half Max (FWHM) of 40 nm. The AS7263 measures six channels in the red to near-infrared (NIR) region (610, 680, 730, 760, 810, and 860 nm) with a FWHM of 20 nm. Meanwhile, the AS7265x consists of three integrated 6-channel sensors, measuring 18 spectral channels from the ultraviolet (UV) to the NIR region (410, 435, 460, 485, 510, 535, 560, 585, 610, 645, 680, 705, 730, 760, 810, 860, 900, and 940 nm), each with an FWHM of 20 nm.

The leaves were removed from the plants, stored in plastic boxes, and immediately taken to the lab, where their spectra were measured. For large banana leaves, a section of the leaf was cut for testing, while long sugarcane and rice leaves were cut off from the stalks. A black background was placed underneath the leaves for the measurements. A white A4 paper was used for the reference spectra. The raw sensor data were then converted to reflectance and absorbance using the following equations:(1)$$R=\frac{R_{\mathrm{measured}}}{R_{\mathrm{ref}}}$$(2)$$A=-\log_{10}\left(R\right)$$

*R* represents the reflectance of the leaf relative to the reference reflectance. $R_{\mathrm{measured}}$ is the reflectance measured from the leaf, $R_{\mathrm{ref}}$ is the reflectance measured from the reference material (white paper), and *A* is the absorbance.

### 2.4. Leaf Chlorophyll Measurements

After spectral measurements were collected, chlorophyll extraction was performed to obtain reference values for model development and validation. Chlorophyll levels were measured following the method described in [53]. Briefly, three disk samples from each leaf were obtained with a standard paper hole puncher and then placed in dimethylformamide (DMF). The chlorophyll concentration of the solutions was measured using a UV–Vis spectrophotometer (Model Specord 40, Analytik Jena, Germany) after 48 h for mango, rice, and sugarcane leaves and 96 h for banana and jasmine leaves due to their increased thickness. The total chlorophyll levels were calculated using the following equation: (3)$$\mathrm{Total}\;\mathrm{Chlorophyll}\;\left(\mu g/\mathrm{mL}\right)=0.18\cdot \left(20.27\cdot A_{647\;\mathrm{nm}}+7.04\cdot A_{664\;\mathrm{nm}}\right)$$
where the absorption coefficients ($A_{x\;\mathrm{nm}}$) in these equations for the specified wavelengths (xnm) were derived from chlorophyll absorption characteristics in DMF, as described in [53]. The factor 0.18 accounts for the path length and the dilution factor of the DMF solution.

Outliers in the reference chlorophyll data were removed using a 3-sigma cutoff. Appendix A provides a detailed description of the reference chlorophyll data and the outlier removal process, including graphical representations of the data and their spread.

### 2.5. Data Processing and Statistical Methods

After collecting spectral and chlorophyll reference measurements, data processing and statistical analysis were performed to fit the spectral data to chlorophyll levels using regression models. All data preparation, fitting, and visualization were conducted using Python, and the scripts for these analyses are available in the analysis folder of the GitHub repository. The Mahalanobis distance was calculated using the following formula: (4)$$D^{2}={\left(x-\mu \right)}^{T}\Sigma^{-1}\left(x-\mu \right)$$
where $D^{2}$ is the Mahalanobis distance, *x* is the individual spectra, μ is the mean spectra, Σ is the covariance matrix, $\Sigma^{-1}$ is the inverse of the covariance matrix, and ${\left(x-\mu \right)}^{T}$ denotes the transpose of the vector (x−μ) [54].

Partial least squares (PLS) regression and learning curves were implemented using the scikit-learn library. The Akaike Information Criterion (AIC) was calculated as follows: (5)$$\mathrm{AIC}=n\cdot \ln\left(\frac{\mathrm{RSS}}{n}\right)+2\cdot k$$
where *n* is the number of observations, RSS is the residual sum of squares, *k* is the number of parameters in the model (number of latent variables), and ln is the natural logarithm.

To evaluate the PLS models, Monte Carlo splitting (Shuffle Split) divided the data into training and test sets. Group-based splitting confined all spectra from a single leaf to either the training or test set, thereby preventing data leakage. All spectra were fit to the PLS model, and the final predicted leaf chlorophyll levels were calculated as the average of the predictions from each spectrum of the leaf. A stratified split was applied to maintain an equal distribution of chlorophyll levels between sets [55]. Chlorophyll levels were quantilized into 10 bins, and each split was sampled equally across these bins.

## 3. Results

### 3.1. Leaf Spectra

The AMS sensors are designed for original equipment manufacturers (OEMs), meaning that users must set the integration time and LED current. The optimal settings were determined using N-way ANOVA analysis, with the procedure detailed in Appendix A. In summary, the AS7262 and AS7265x performed best with the shortest integration time of 140 ms, while the AS7263 achieved optimal performance with the longest integration time of 700 ms. The lowest current setting (12.5 mA) was found to be optimal for all sensors. These settings were selected for further analysis.

Figure 4 shows the leaf spectra, taken in triplicate, of mango leaves measured by all three sensors. The left column (Figure 4a–d) shows the raw, unnormalized light intensity readings from the sensors, while the right column (Figure 4e–h) displays the reflectance normalized to the reference, a white sheet of A4 paper. The x-axis is shared across all figures to compare the spectral regions measured by each sensor. The black lines represent the mean spectra for each leaf–sensor combination. Each spectrum is color-coded to illustrate the chlorophyll level of the associated leaf, according to the color bar on the right. Leaves with lower chlorophyll levels have a higher overall reflectance due to lower chlorophyll content, a thinner outer-surface cuticle, and less developed cell structures [56].

Figure 4a,e display the AS7262 spectra with illumination provided by a white LED. The spectra exhibit the characteristic green color of chlorophyll, where the blue and red regions of the spectra reflect less light compared to the green region in both the raw intensity (Figure 4a) and the normalized reflectance (Figure 4e). Figure 4b,f display the AS7263 spectra, also illuminated by a white LED. The raw light intensity (Figure 4b) shows a peak in the red region at the 600 nm channel and a small near-infrared (NIR) component, as the illuminating LED only has a small NIR signal. The normalized reflectance (Figure 4f) reveals the characteristic leaf reflectance pattern, with low red reflectance but a large NIR reflectance.

Figure 4c,g present the spectra from the AS7265x sensor illuminated by a white LED. The raw light intensity (Figure 4c) exhibits a distinctive spiky pattern with three peaks in the visible region. The first peak at the 460 nm channel arises because the white LED uses a blue LED with a phosphorus coating to produce white light. According to the LED datasheet, the blue LED emits a sharp peak around 455 nm, which causes the first peak in the raw intensity readings. The LED then has a smoother spectrum between 500 and 650 nm.

While the first peak is caused by the illumination source, the second and third peaks are due to the sensor’s geometry. The AS7265x consists of three chips: the AS72651, AS72652, and AS762653. The first six channels are on the AS71653 chip, located close to the white LED on the SparkFun breakout board. The next two wavelengths in the spectra, at 560 nm and 585 nm, are on the AS72652 chip, which is located further from the LED, decreasing the light intensity. The peaks at 535 and 610 nm are not caused by those channels having higher intensities but by the intensity in the middle two channels decreasing. The 645 nm channel decreases in intensity for two reasons: first, leaf reflectance is lower in this region, and second, this channel is on the AS72652 chip, which again is farther away from the LED. Finally, the NIR region of the spectra is low because the white LED has low intensity in this region.

The normalized reflectance of the AS7265x (Figure 4g) shows a typical leaf reflectance pattern, with low intensity in the red and blue regions, moderate reflectance in the green region, and the highest reflectance in the NIR region. However, in the deeper IR regions (>800 nm), there is reduced reflectance and a noisier signal because the white LED is not an ideal light source for this region. To better measure IR reflectance, the IR LED on the SparkFun breakout board was activated with the white LED, and its spectra were recorded, as shown in Figure 4d,h. These spectra follow a similar pattern as those in Figure 4c,g but with a larger raw intensity and less noisy reflectance in the IR region from 810 nm to 940 nm.

### 3.2. Outlier Detection

Next, the spectra were examined for errors in reflectance readings. One popular method for outlier detection is to measure the Mahalanobis distance between spectra and remove points with large distances. Unlike Euclidean distance, which considers all points in the spectrum equally, Mahalanobis distance accounts for correlations within the spectra before calculating the distance. This is relevant for leaf spectra, where the six NIR channels of the AS7265x are highly correlated. A slight change in the NIR region could cause an oversized change in Euclidean distance but not in Mahalanobis distance.

Figure 5 displays the spectra for mango leaves, similar to Figure 4, but with outliers highlighted in red for the three sensors. Figure 5a,c,e identifies outliers based on a 3-sigma cutoff for Mahalanobis distance calculated from the spectral data. For the AS7262 (Figure 5a) and AS7265x (Figure 5e) sensors, 6–7 readings are identified as outliers due to their high reflectance in the visible region, a trend seen in both sensors. Additionally, the outliers measured with the AS7265x showed low NIR reflectance. These spectral characteristics suggest the outliers are young, undeveloped leaves, which typically have high visible reflectance due to low chlorophyll content and a thinner cuticle, as well as lower NIR reflectance due to an underdeveloped spongy mesophyll layer [57].

The identified outliers were from two leaves (and one additional spectrum for the AS7265x) that were not fully developed. These leaves were intentionally oversampled, not randomly selected, to obtain a wide range of chlorophyll levels for modeling. While the large Mahalanobis distances classified these spectra as outliers, this result is from the oversampling strategy to obtain a broader range of chlorophyll content rather than measurement errors.

To identify truly erroneous readings, Mahalanobis distances were calculated from the residue of each spectrum, where the residue is the difference between an individual spectrum and the average of the other two spectra from the same leaf [58]. Outliers were then identified as readings with distances exceeding 3-sigma from the average. For mango leaves, no outliers were identified for the AS7262 and AS7265x sensors using this method. This indicates that the previously identified outliers were due to the spectral properties of young leaves, not sensor errors.

Outliers identified based on residue Mahalanobis distances were removed, and the remaining spectra were used for further analysis. A total of 31 recordings out of 4500 were removed, with half from the AS7263 sensor, probably due to the reduced signal-to-noise ratio caused by using a white LED with an NIR sensor. These results demonstrate the stability of the sensors for several months, as 99% of all readings were deemed acceptable.

Figure 6 shows the relative reflectance of the remaining four leaves measured by the three sensors, with the removed outliers shown in red. Banana leaves, shown in Figure 6a–c, display trends similar to mango leaves but with a narrower reflectance range, indicating more consistent and uniform reflectance properties. This difference stems from the different development of the leaves, and the sampling processes for the two plants. Mango leaves, typical of trees, emerge as light-colored buds before maturing. Some of these undeveloped leaves were included in the measurements, as seen in the mango spectra, where a few samples show green reflectance levels greater than 35%. Bananas, on the other hand, are large herbaceous plants that produce leaves that develop rolled up within the pseudostem. Therefore, these undeveloped banana leaves are inaccessible for measurement, resulting in a narrower range of variability in the banana spectra.

Jasmine leaves show a similar pattern in Figure 6d–f, with a few nuances. First, jasmine leaves have a narrower chlorophyll range compared to the other species tests. This would typically result in a smaller reflectance range. However, jasmine leaves are thicker, rougher, and more textured than mango or banana leaves. These differences stem from the structural characteristics of jasmine leaves, which include striated cuticles with folds, denser mesophyll layer (internal tissue where photosynthesis occurs), and undulating epidermal cell walls [59]. The combination of a narrower chlorophyll range and a rougher texture results in a reflectance spread between leaves that is similar to that of banana leaves.

Rice (Figure 6g–i) and sugarcane (Figure 6j–l) leaves have a wider range of reflectance values, with many spectra deviating from the average. This variation is likely due to their narrow shape, which prevents some leaves from fully covering the optical path of the sensors. As a result, the spectra may include varying contributions from the leaves and the black background. This effect is more pronounced for the AS7265x, which has a larger shroud due to its size, and in rice leaves that are narrower than sugarcane leaves. The inclusion of these narrower leaves in this study aims to evaluate the sensor performance without the need to confine the optical path using light pipes or other complex arrangements.

### 3.3. Model Selection

After collecting the chlorophyll and spectra data, a regression model was developed to predict chlorophyll levels from the spectra. Partial least squares (PLS) regression was selected because it effectively handles correlations between spectral channels by projecting the data onto latent variables (LVs) that are orthogonal (uncorrelated). This approach is similar to principle component analysis (PCA); however, while PCA identifies principle components that maximize the variance in the predictors (spectra) alone, PSL selects LVs that maximize the co-variance between the predictors and the response variable (chlorophyll levels).

To optimize a PLS model, the number of LVs must be carefully selected to balance model complexity and predictive accuracy. Increasing the number of LVs will increase a model’s fit to the training data and possibly to the test data, but it can lead to overfitting, reducing its ability to generalize to new data. The Akaike Information Criterion (AIC) was used to determine the optimal number of LVs. The AIC considers the model’s goodness of fit, the number of data points, and complexity (in this case, the number of LVs) [60]. A lower AIC score indicates a better balance between the model’s goodness of fit and complexity.

To calculate the AIC score and evaluate the model, a double-nested cross-validation (CV) approach was used. The outer CV loop split the dataset into a training set (80 leaves) to select the number of LVs and a validation set (20 leaves) to evaluate the model. The inner CV loop further divided the training set into 60 leaves for training the PLS model and 20 leaves as a test set to calculate the AIC and $R^{2}$ scores. The number of LVs that minimized the AIC score was selected. Since the number of LVs is chosen based on the inner test set (80 leaves), the model incorporates information from both the inner training and test sets. Therefore, an independent outer validation set, not used in the inner CV, is required to evaluate the model’s $R^{2}$ and mean absolute error (MAE).

Figure 7 shows the relationship between the number of LVs in the PLS model and the test $R^{2}$ scores (in orange) and AIC scores (in green) for mango leaves. The standard deviations are shown as shaded regions. The inner CV was run 20 times for each number of LVs to obtain reliable estimates for the optimal number of LVs. The selected model was then evaluated on the validation set. The outer CV was run 50 times to estimate validation $R^{2}$ and MAE scores. In total, the inner CV loop ran 1000 iterations for each number of LVs, and the average results are plotted in Figure 7. The same procedure was used to find the optimal number of LVs for each of the other four leaves for each sensor. The scans are plotted in Appendix A, and the optimal number of LVs for all combinations is summarized in Table 1. Most plots were similar, except for the high noise level in the rice leaf curves.

### 3.4. Validation Scores

The heatmaps in Figure 8 display the $R^{2}$ and MAE values for the outer CV validation scores from the PLS parameter scans. As expected, the AS7265x sensor, with a larger number of channels, had the best fit for all leaves except rice, which will be discussed later. The AS7265x showed MAE values around 3.5 $\mu {g/\mathrm{cm}}^{2}$, and $R^{2}$ scores ranging from 0.96 to 0.81, for leaves other than rice. AS7262 and AS7263 performed similar to each other with MAE values between 3.4 and 5.6 $\mu {g/\mathrm{cm}}^{2}$ and $R^{2}$ scores in the range of 0.93–0.73, with a large performance difference between leaves. The performance of each individual leaf species will be discussed in the next section.

### 3.5. Results by Leaf Species

#### 3.5.1. Mango Leaves

Figure 9 presents the results of fitting the spectral data to PLS models. Figure 9a–c show actual versus predicted chlorophyll levels for a single test (green) and training (purple) split for the AS7262 (Figure 9a), AS7263 (Figure 9b), and AS7265x (Figure 9c) sensors. Figure 9d–f shows the learning curves that plot the MAE for the testing and training sets as a function of the number of leaves used to train the PLS model.

The AS7262’s $R^{2}$ scores for the training, test, and validation sets are 0.93, 0.93, and 0.92, respectively, with corresponding MAE scores of 4.85, 5.11, and 5.41 $\mu {g/\mathrm{cm}}^{2}$, respectively. The slightly lower validation score arises because the PLS scans occasionally select a different number of LVs for certain splits, leading to slightly less optimal fits. Although five LVs best fit the current dataset, four and six LVs were sometimes selected in the PLS scans as they yielded similar, though somewhat worse, results. Variability in future datasets, from noise or different spectral characteristics of the leaves, may result in a different optimal number of LVs. In such cases, using the current PLS model with 5 LVs may slightly reduce test scores. However, because the validation score accounts for model generalization and unseen data, it is more representative of future measurements. Most reported scores in the literature, though, are based on the test set or sometimes even the training set, which may overestimate the model’s generalizability. This difference should be small for large datasets, as seen in this case.

PLS has only one hyperparameter to tune (the number of LVs) and isolates noise into a smaller number of higher-order LVs, minimizing the difference between validation and test scores. In contrast, non-linear regressors with multiple hyperparameters often exhibit larger differences, with test scores being significantly better than validation scores [61].

The AS7263 exhibited lower performance in fitting chlorophyll levels, with $R^{2}$ scores of 0.92, 0.91, and 0.90 for the training, test, and validation sets, respectively. The decrease in accuracy may result from the measured spectral region being less optimal for measuring leaf chlorophyll levels or from a mismatch between the illuminating LED spectrum and the sensor’s measurement range. Such a mismatch reduces the signal-to-noise ratio, necessitating longer integration times to compensate. However, longer integration times cause thermal drift, lowering measurement accuracy, see Appendix A.

To evaluate the impact of these factors, the AS7262 test scores were calculated using the same integration time of 700 ms as AS7263. The longer integration time reduced the AS7262 test $R^{2}$ score to 0.91, the same as the AS7263 score. This suggests that the two sensors could perform similarly if the breakout board was equipped with an IR LED matched to the AS7263’s spectrum. The AS7263 learning curves showed the slowest convergence of the test and training sets, with convergence still occurring at 80 leaves in the training set.

The AS7265x demonstrated the best performance, with validation and test $R^{2}$ scores of 0.96 and a training score of 0.98. The learning curves indicated that approximately 48 leaves in the training set were sufficient for the test and training set scores to level off. However, the learning curves revealed a persistent gap between the test and training scores rather than convergence. This discrepancy is caused by the unique geometry of the AS7265x sensor, a factor discussed in the section on rice leaves, where this effect is more pronounced and easier to observe.

#### 3.5.2. Banana Leaves

Figure 10 presents the results of fitting the spectral data from banana leaves to chlorophyll levels. The AS7262 performed slightly worse on banana leaves compared to mango leaves, while the AS7263 showed slightly better performance. This difference could be due to the red/NIR region being more effective than the visible region for thicker leaves such as banana leaves. The learning curves for the AS7263 indicated slower convergence, requiring around 60 leaves, compared to the AS7262, which converged around 32 leaves.

The AS7265x fit banana leaves similar to mango leaves, with a validation $R^{2}$ score of 0.95, a test $R^{2}$ score of 0.96, and a validation MAE of 3.52 $\mu {g/\mathrm{cm}}^{2}$. Its learning curves showed that test and training scores leveled off around 45 leaves, with a persistent gap noted before.

#### 3.5.3. Sugarcane Leaves

Figure 11 shows that the $R^{2}$ scores for all sensors were lower for sugarcane leaves compared to mango or banana leaves. This may be due to interference from the prominent midrib and the rough texture of sugarcane leaves, which have prominent trichomes (hair-like structures). In contrast, mango and banana leaves are uniform and smooth, allowing for more consistent measurements. This highlights a limitation of the AMS sensors, which require a larger measurement area compared to devices like the SPAD meter. For narrower leaves, such as sugarcane, the larger measuring area can necessitate including the non-uniform leaf features, such as a midrib, in the measurement, unlike smaller sensors that can measure just the leaf blade. Additionally, some young leaves did not fully cover the optical path, potentially reducing measurement accuracy.

Similar to banana leaves, the AS7263 slightly outperformed the AS7262 in both $R^{2}$ and MAE scores, while the AS7265x had the best fit to chlorophyll levels. Notably, although the $R^{2}$ scores were worse than those for mango and banana leaves, the MAE values were better. The learning curves indicated that all sensors were optimally fit by 32 leaves in the training set.

#### 3.5.4. Jasmine Leaves

Jasmine leaves were selected to examine how different leaf morphology affects the measurement of chlorophyll levels, as they are less uniform in thickness and have prominent venation. Figure 12 shows that, as expected, all sensors had lower (worse) $R^{2}$ scores compared to mango or banana leaves, although MAE values were lower (better). This is due to the narrower range of chlorophyll levels in jasmine leaves, highlighting how leaf choice and chlorophyll range strongly influence $R^{2}$ scores. Notably, the MAE of the reference measurements relative to the leaf average was 1.96 $\mu {g/\mathrm{cm}}^{2}$, as seen in Appendix A, which will lower the accuracy of fitting chlorophyll levels [62]. Across all sensors, the learning curves showed that the models were optimally fit using 32–40 leaves in the training set.

#### 3.5.5. Rice Leaves

Figure 13 presents the fitting results for rice leaves, which are narrower than the other leaves tested and pose a challenge for AMS sensors to accurately fit chlorophyll levels due to the large measurement area required for reflection measurements. The learning curves in Figure 13d–f highlight these challenges, with the test and training scores for the AS7265x failing to converge even with 80 leaves in the training set. In comparison, the AS7263 converged at 80 leaves, while the AS7262 converged at 72 leaves.

This abnormal behavior in the learning curves is accompanied by two other anomalies. First, the AS7265x $R^{2}$ scores for the validation, test, and training sets are 0.75, 0.86, and 0.94, respectively, showing a much larger disparity than in other sensor–leaf combinations. Second, rice leaves are the only leaf type where the AS7265x had a lower validation score than the AS7262 and AS7263 despite its broader measurement range. A possible explanation for these observations is that the three separate sensors of the AS7265x (AS72651, AS72652, AS72653) have different apertures, potentially causing interference when combining different sensor outputs on non-uniform surfaces. To investigate whether the separate sensors cause these issues, each of the individual sensors in the AS7265x was fit with a PLS model using 5 LVs, the average optimal number determined for the AS7262 and AS7263 sensors.

Figure 14 presents the test scores of fitting chlorophyll levels for all four conditions: the three individual chips and the combined output. For all leaves except rice, the individual chips performed worse than the combined sensor, as expected. However, for rice leaves, the single AS72652 chip outperformed the combined sensor, implying that combining sensors with different apertures can reduce performance when measuring non-uniform surfaces. To overcome this, selecting wavelengths from a single chip, or using a setup to confine the optical path [63,64], can improve performance.

## 4. Discussion

### 4.1. Sensor Setup and Characterization

The AMS spectral sensors (AS7262, AS7263, and AS7265x) were tested to evaluate their ability to measure the chlorophyll levels of five species of leaves. The AS7262 has six spectral channels in the visible range, the AS7263 has six channels in the red/NIR region, and the AS7265x has 18 channels spanning the visible to NIR spectrum. The sensors were tested on SparkFun breakout boards that included LEDs for illumination. All sensors are equipped with a white LED, while the AS7625x includes UV and IR LEDs. The breakout boards were connected to a SparkFun Arduino-compatible microcontroller through a Qwiic multiplexer. The microcontroller code to operate the sensor and a GUI for data collection are available at https://github.com/KyleLopin/asm_chloro_test/tree/master/source (accessed on 25 March 2025).

Each sensor has a current sink pin to drive the illumination LEDs. Through electrical measurements and chlorophyll data fitting, 12.5 mA was determined to be the optimal current level for all sensors (see Appendix A). This is likely due to heat generated by the sensors while sinking higher currents, which can reduce spectral accuracy. Different integration times were tested, and it was found that the AS7262 and AS7265x performed best at the shortest tested integration time (140 ms). This is likely due to thermal drift increasing at longer times, as the sensor sinks the LED current for a longer duration. In contrast, the AS7263 performed best at the longest integration time tested (700 ms), likely due to the lower signal level in its NIR measuring range provided by the breakout board’s white LED.

### 4.2. Data Preparation and Model Selection

Reference leaf chlorophyll measurements were made in triplicate using standard extraction methods, and outliers were removed from the data using a 3-sigma cutoff. The data and results for this procedure are shown in Appendix A.

The sensors collected spectra from the leaves, and spectral outliers were removed using a 3-sigma cutoff based on the Mahalanobis distance of the residues of each individual spectrum relative to the other two spectra from the same leaf. This method effectively removed outliers while preserving samples with unique features, such as those found in newly formed leaves, compared to calculating the Mahalanobis distance to the full spectral dataset.

Partial least squared (PLS) models were used to fit the spectra to reference chlorophyll levels. The optimal number of latent variables (LVs) was determined using the Akaike Information Criterion (AIC), which balances goodness of fit with model complexity [65]. A nested cross-validation approach was employed to evaluate the model and prevent data leakage. The dataset was split into validation, test, and training subsets. The training set was used to generate the PLS model, the test set was used to calculate the AIC score and select the optimal number of LVs, and the validation set was used to evaluate the model performance and calculate the $R^{2}$ and MAE scores.

### 4.3. Sensor Performance

Figure 8 presents heatmaps of the $R^{2}$ and MAE values for the validation set, illustrating the performance of each sensor across different leaf species. Each sensor exhibited a spread between different leaf species of 2 $\mu {g/\mathrm{cm}}^{2}$ in MAE and 0.2 in $R^{2}$, highlighting variability in performance between species. Additionally, N-way ANOVA results in Appendix A showed that leaf species statistically affected the results, with *p*-values smaller than the tolerance level.

The variability in Figure 8 aligns with findings from previous studies, which also observed significant differences in chlorophyll measurement accuracy across species. For example, the $R^{2}$ between atLEAF CHL PLUS sensor readings and chlorophyll levels was 0.95 for beech trees but only 0.83 for silver birch trees [66]. Similarly, $R^{2}$ scores for shrubs ranged from 0.82 for honeysuckle to 0.34 for buckthorn [67]. These patterns are consistent with the current results, showing higher $R^{2}$ scores for chlorophyll models for tree leaves compared to shrubs.

Additionally, a study using AS7262 and AS7263 sensors in combination reported a similar range of $R^{2}$ values for nitrogen content, from 0.63 for wheat to 0.82 for soybeans [39]. These findings underscore that the performance of chlorophyll sensors cannot be judged in isolation, but the species of leaves being measured should also be considered.

To put the sensors performances on mango leaves, with MAEs of 3.6–5.6 $\mu {g/\mathrm{cm}}^{2}$, in perspective, different mango cultivars within the same season can have average chlorophyll levels ranging from 91 to 50 $\mu {g/\mathrm{cm}}^{2}$, while the same cultivar can exhibit seasonal variations from 91 to 31 $\mu {g/\mathrm{cm}}^{2}$ [68]. Typical mango plantations yield around five metric tons per hectare, but well-managed plantations can achieve yields of 20–30 metric tons per hectare, emphasizing the need for tools to support optimal management practices [69]. Previous studies have reported $R^{2}$ scores of 0.91 for correlating SPAD measurements to chlorophyll levels and 0.94 for a high-resolution spectrometer covering a range from 282 to 1097 nm [70]. These findings, together with the current results, highlight that the uniform and smooth surface of mango leaves allows for accurate chlorophyll measurement, with AMS sensors showing comparable or slightly improved performance over previous reports.

Previous research on banana leaves reported an $R^{2}$ of 0.86 for measuring chlorophyll levels with a SPAD meter [71] and 0.65 for a UAV-mounted camera [72]. All sensors demonstrate sufficient accuracy for monitoring the health and development of banana plants, as Mg^2+^ deficiencies have been shown to cause a 50% reduction in chlorophyll levels [73]. Additionally, developmental studies indicate that chlorophyll levels in banana leaves increase by 35% between 90 and 150 days after planting, while different fertilizer treatments can lead to a 65% increase in chlorophyll levels [74].

Previous studies on correlating sugarcane leaf chlorophyll levels with SPAD readings have reported $R^{2}$ scores ranging from 0.92 to 0.65 depending on the plant’s growth stage [75,76]. Additionally, the FieldSpec 3 spectroradiometer achieved $R^{2}$ scores ranging from 0.77 to 0.61, which was accurate enough to distinguish between different fertilizer treatments [77]. Chlorophyll levels in sugarcane leaves have also shown a strong correlation with yield ($R^{2}$ = 0.87) [75]. The performance of all three AMS sensors in this study was comparable to or better than previous reports, with the AS7265x performing the best of the AMS sensors.

Rice leaves were best fit by the AS7262 sensor, with an $R^{2}$ score of 0.88 and a MAE of 4.72 $\mu {g/\mathrm{cm}}^{2}$. The full AS7265x sensor performed worse on rice leaves, though if the channels of the single AS72652 sensor were used, a better $R^{2}$ score of 0.86 and a MAE of 4.51 $\mu {g/\mathrm{cm}}^{2}$ were achieved. This is comparable to previous reports of fitting SPAD measurements to chlorophyll levels. Among 22 species tested using the SPAD meter, rice had the lowest $R^{2}$ score of 0.82 [78]. Another study found that, across two cultivars and three stages of rice growth, SPAD $R^{2}$ scores ranged from 0.71 to 0.86 [79].

It has also been reported that using the destructive measurement method, even within the same leaf layer, the standard deviation of chlorophyll levels was approximately 5 $\mu {g/\mathrm{cm}}^{2}$, which is similar to the error reported by all AMS sensors [80]. The same study showed that the difference in chlorophyll levels between rice without fertilizer and with 240 kg ha^−1^ nitrogen fertilizer added was 30–40 $\mu {g/\mathrm{cm}}^{2}$ (for the fourth leaf upon emergence). Additionally, chlorophyll levels in leaves without nitrogen fertilizer applied decreased from 44 to 28 $\mu {g/\mathrm{cm}}^{2}$ between the V1 and R1 stages (approximately 7 to 50 days after germination), while chlorophyll levels remained stable when nitrogen fertilizer was applied [80].

Site Specific Nitrogen Management (SSNM) studies frequently use SPAD readings, including one study that varied nitrogen application based on SPAD readings of >40, 40–35, and <35 [81]. Using previously published SPAD to chlorophyll conversion equations, this corresponds to >63.6, 63.6–57.5, and <57.5 $\mu {g/\mathrm{cm}}^{2}$ [78]. This range is close to the MAE of the AMS sensors and within the variability of chlorophyll levels within the same leaf layer. More research is needed to determine whether AMS sensors can be used effectively for SSNM. However, their comparable performance to SPAD meters is promising.

### 4.4. Overall Performance and Limitations

A key distinction between AMS sensors and commercial chlorophyll meters like SPAD is their need for additional hardware and programming. Unlike standalone SPAD meters, AMS sensors require a microcontroller and dedicated software, which limits their ease of use. Although AMS sensor-based devices have been reported for specific applications, future work is needed to develop open-source devices. This would also enable the integration of additional features such as IoT, AI, and edge computing if desired.

Additionally, sensor design affects accuracy, as the AS7265x consists of three separate sensors, each with its own aperture, which introduces inconsistencies when measuring non-uniform surfaces, such as narrow leaves. This limitation may reduce its accuracy compared to single-aperture sensors like the AS7262 and AS7263. However, optical diffusers may help mitigate this by homogenizing the light entering each sensor, though this would increase both the complexity and cost of the device.

As demonstrated in this report, calibration models must also be developed to correlate sensor output with chlorophyll levels, which are determined using a destructive method. The learning curves presented in this report provide guidance on how many leaves are required for calibration. For example, Figure 11 shows that the AS7262 sensor can estimate chlorophyll levels in sugarcane leaves with an error of 4 $\mu {g/\mathrm{cm}}^{2}$ using a calibration model with 32 leaves, equivalent to around one week of data collection. Once calibrated, these sensors can rapidly and efficiently measure hundreds to thousands of other leaves.

Despite these limitations, the low cost and reasonable accuracy of AMS sensors make them a potentially valuable tool for agricultural applications. The AS7262, priced at just USD 28, offers moderate accuracy, making it ideal for cost-sensitive applications, such as education or recreational use. The AS7265x, costing USD 70, provides better accuracy, offering comparable or better performance than the more expensive SPAD meter (USD 1000) and atLEAF meter (USD 250). However, further development of open source devices is still needed to improve usability. This combination of affordability and accuracy highlights the potential of AMS sensors for precision agriculture, particularly for small-scale farmers who require cost-effective solutions to optimize inputs like fertilizers and improve crop yields.

## 5. Conclusions

The low-cost AMS sensors (AS7262, AS7263, and AS7265x) were used to fit chlorophyll levels using partial least squares (PLS) regression. To ensure robustness and prevent data leakage, the models were evaluated using the validation set from an outer nested cross-validation approach. The AS7265x, a collection of three 6-channel sensors, demonstrated good performance on smooth, uniform leaves such as those of banana and mango, with $R^{2}$ scores of 0.95–0.96. For non-uniform leaves, jasmine, sugarcane, and rice, the AS7265x achieved $R^{2}$ scores ranging from 0.75 to 0.85. The low-cost 6-channel AS7262 and AS7263 sensors achieved $R^{2}$ scores between 0.93 and 0.86 for smooth, uniform leaves, and 0.85–0.73 for the other leaves. These results demonstrate that affordable AMS sensors can reliably measure chlorophyll levels across various leaf types, with performance varying based on leaf morphology, making them a promising tool for precision agriculture applications such as optimizing nutrient management and monitoring crop development.

## Figures and Tables

**Figure 1 sensors-25-02198-f001:**
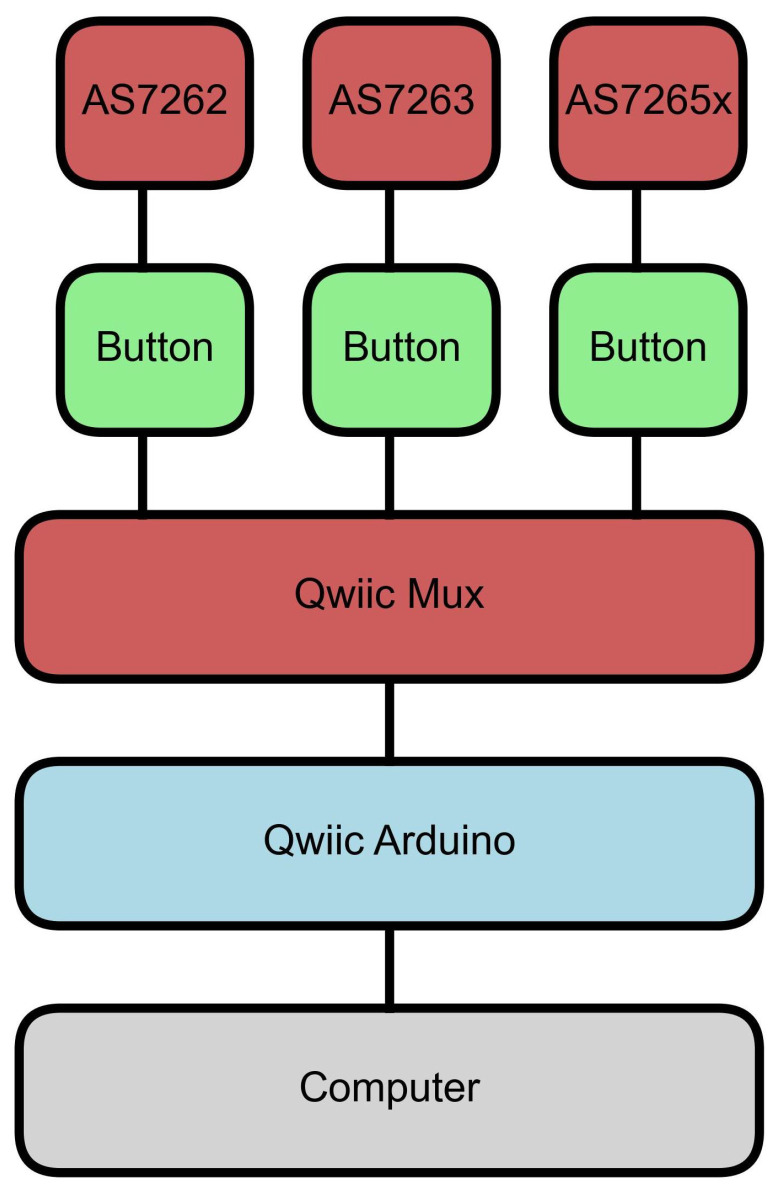
Diagram of device setup. A Qwiic button was connected to each of the sensors, AS6272, AS7263, and AS7265x. All sensors were then connected to a Qwiic Mux, which was connected to an Arduino with a Qwiic cable. Finally, the Arduino was connected to a computer.

**Figure 2 sensors-25-02198-f002:**
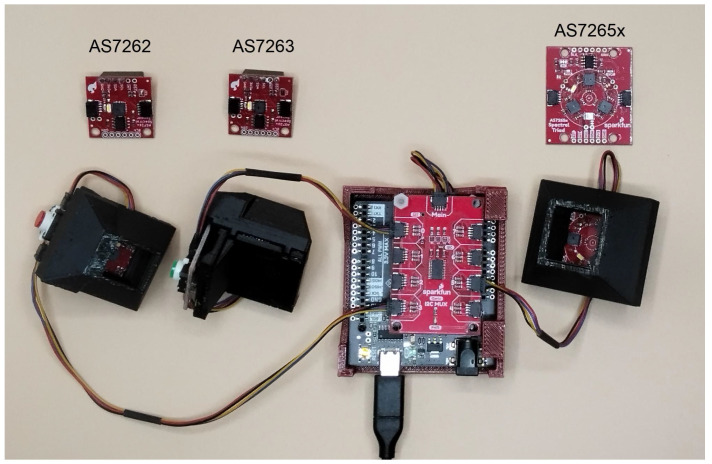
Device setup. Photograph of the AMS sensors (top, labeled) and the sensors mounted in black 3D printed holder/shrouds, connected to buttons, which are then connected to the Qwiic Mux (red PBC in the center). The Qwiic Mux is then connected to an Arduino-compatible Artemis RedBoard (SnowBoard Edition, black PCB below the Mux).

**Figure 3 sensors-25-02198-f003:**
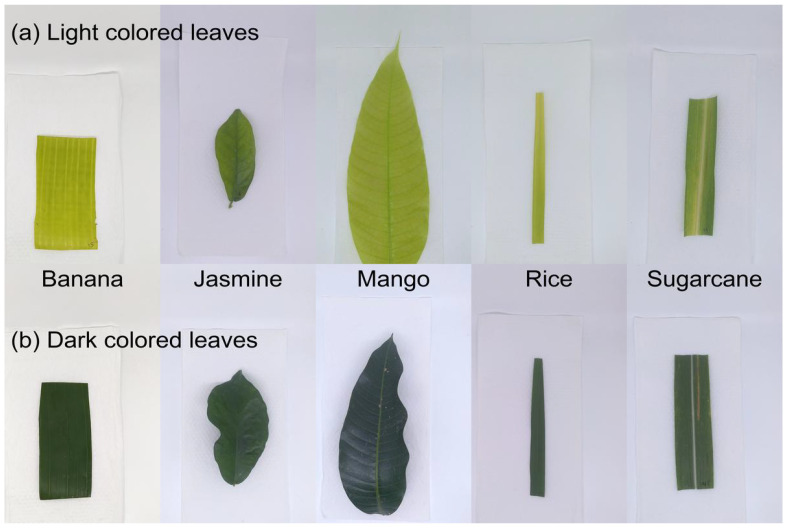
Photographs of leave samples. Photographs of leaf samples arranged in two rows: (**a**) the top row shows light-colored leaves, which are younger, newly emerging leaves, and (**b**) the bottom row shows darker-colored leaves, which are older leaves. The samples for banana, jasmine, mango, rice, and sugar cane are arranged from left to right.

**Figure 4 sensors-25-02198-f004:**
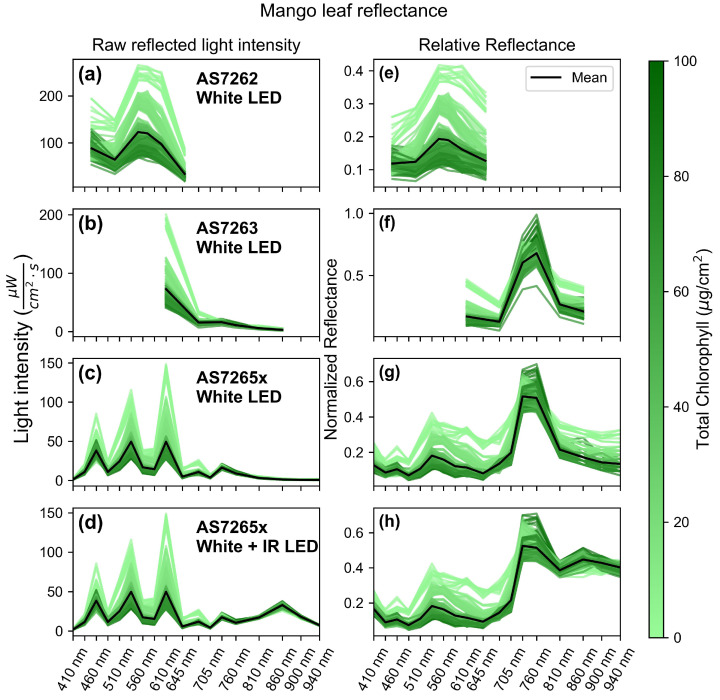
Raw and relative reflected light intensity of mango leaves. (**a**–**d**) Raw light intensity measured by the sensors: (**a**) AS7262, (**b**) AS7263, and (**c**,**d**) AS7265x, all illuminated with a white LED, with (**d**) also using an IR LED. (**e**–**h**) Normalized reflectance relative to a white sheet of paper for the sensors: (**e**) AS7262, (**f**) AS7263, and (**g**,**h**) AS7265x, all illuminated with a white LED, with (**h**) also using an IR LED. The x-axis is uniform for all graphs to compare each sensor’s measurement range.

**Figure 5 sensors-25-02198-f005:**
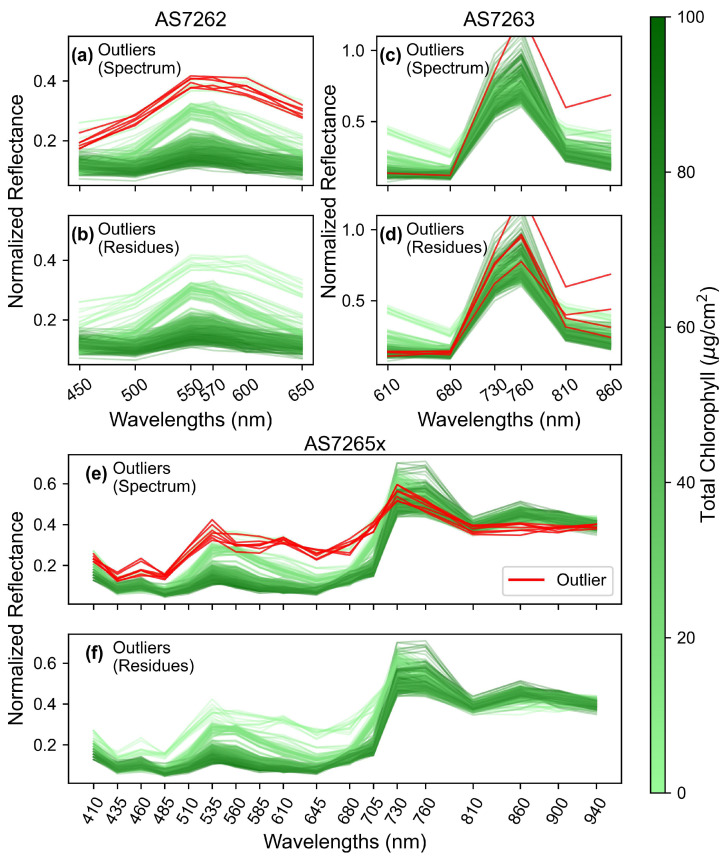
Outlier detection using Mahalanobis distances. (**a**,**c**,**e**) Outliers identified, in red, from the original spectrum. (**b**,**d**,**f**) Outliers identified, in red, from the residue spectrum, calculated as the difference between each individual measurement and the average of the other two measurements on the same leaf; for the sensors: (**a**,**b**) AS7262, (**c**,**d**) AS7263, and (**e**,**f**) AS7265x. Outliers are shown in red, and non-outliers are shown in green, color-coded by chlorophyll levels as indicated by the color bar. The x-axis labels indicate the spectral channels for each sensor.

**Figure 6 sensors-25-02198-f006:**
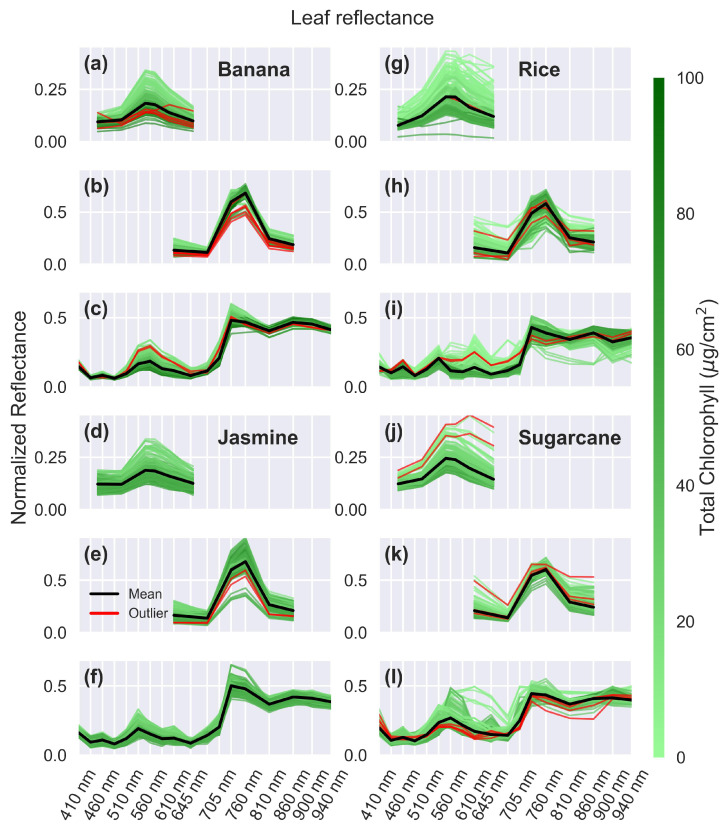
Reflectance with outliers. Relative reflectance for leaves: (**a**–**c**) banana, (**d**–**f**) jasmine, (**g**–**i**) rice, and (**j**–**l**) sugarcane, measured using the sensors AS7262 (**a**,**d**,**g**,**j**) with a white LED, AS7263 (**b**,**e**,**h**,**k**) with a white LED, and AS7265x (**c**,**f**,**i**,**l**) with a white and IR LED. Outliers are shown in red, individual spectra are color-coded according to their chlorophyll levels, and the mean spectrum is shown in black. The x-axis remains consistent across all graphs, with grid lines indicating the AS7265x spectral channels.

**Figure 7 sensors-25-02198-f007:**
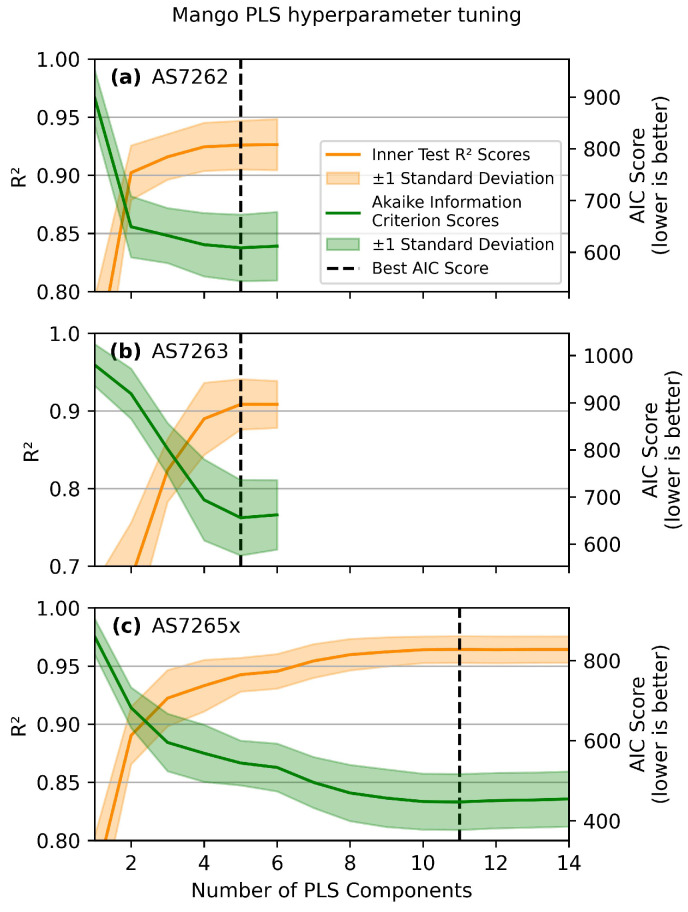
Choosing the number of LVs. PLS hyperparameter tuning scans of mango leaves for the sensors: (**a**) AS7262, (**b**) AS7263, and (**c**) AS7265x, based on the minimum AIC score (green line). Inner CV $R^{2}$ test scores are in orange, with standard deviations as shaded regions, and the minimum AIC score is indicated with the vertical dashed black line.

**Figure 8 sensors-25-02198-f008:**
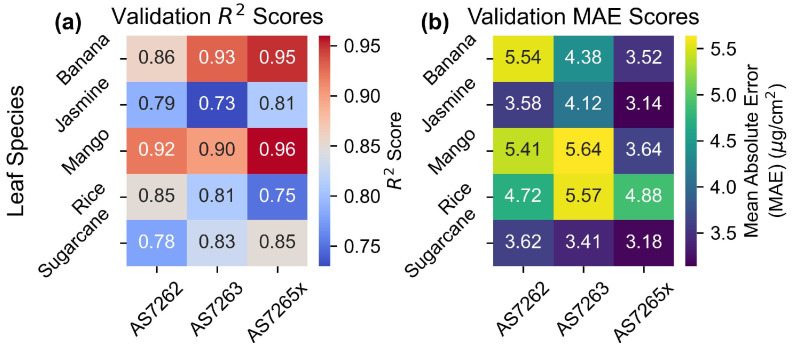
Validation scores. Outer CV validation for (**a**) $R^{2}$ and (**b**) MAE scores.

**Figure 9 sensors-25-02198-f009:**
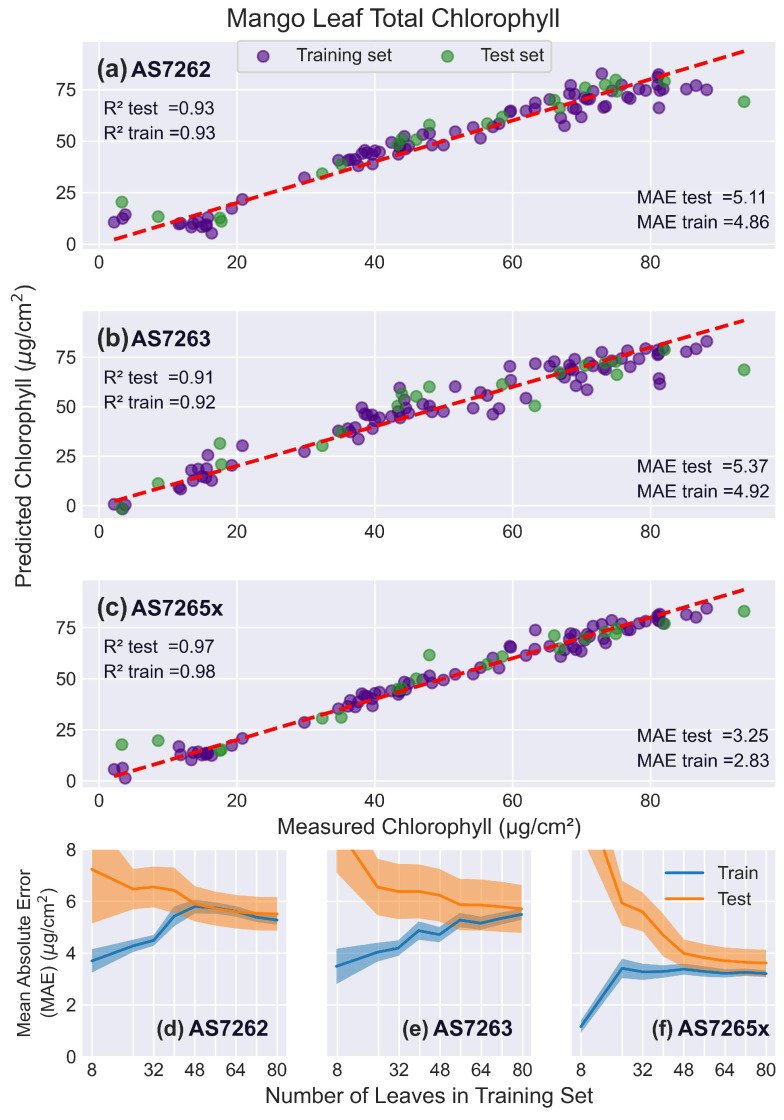
Mango leaf results. PLS model fits for mango leaves. (**a**–**c**) Predicted versus actual chlorophyll levels for the sensors: (**a**) AS7262, (**b**) AS7263, and (**c**) AS7265x, with the red dashed line representing a perfect fit. (**d**–**f**) Learning curves showing the MAE as a function of the number of leaves in the training set, for the sensors: (**d**) AS7262, (**e**) AS7263, and (**f**) AS7265x. Training scores are shown in blue, and test scores in orange, with shaded areas representing the standard deviation.

**Figure 10 sensors-25-02198-f010:**
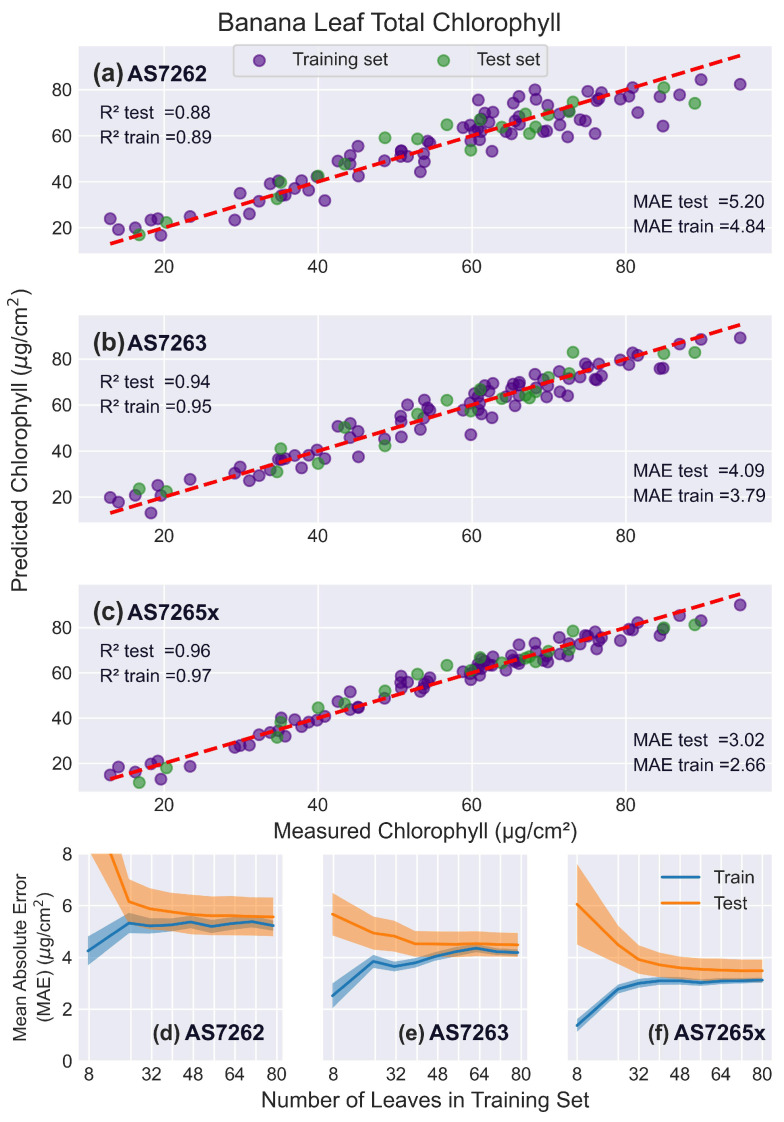
Banana leaf results. PLS model fits for banana leaves. (**a**–**c**) Predicted versus actual chlorophyll levels for the sensors: (**a**) AS7262, (**b**) AS7263, and (**c**) AS7265x, with the red dashed line representing a perfect fit. (**d**–**f**) Learning curves showing the MAE as a function of the number of leaves in the training set, for the sensors: (**d**) AS7262, (**e**) AS7263, and (**f**) AS7265x. Training scores are shown in blue, and test scores in orange, with shaded areas representing the standard deviation.

**Figure 11 sensors-25-02198-f011:**
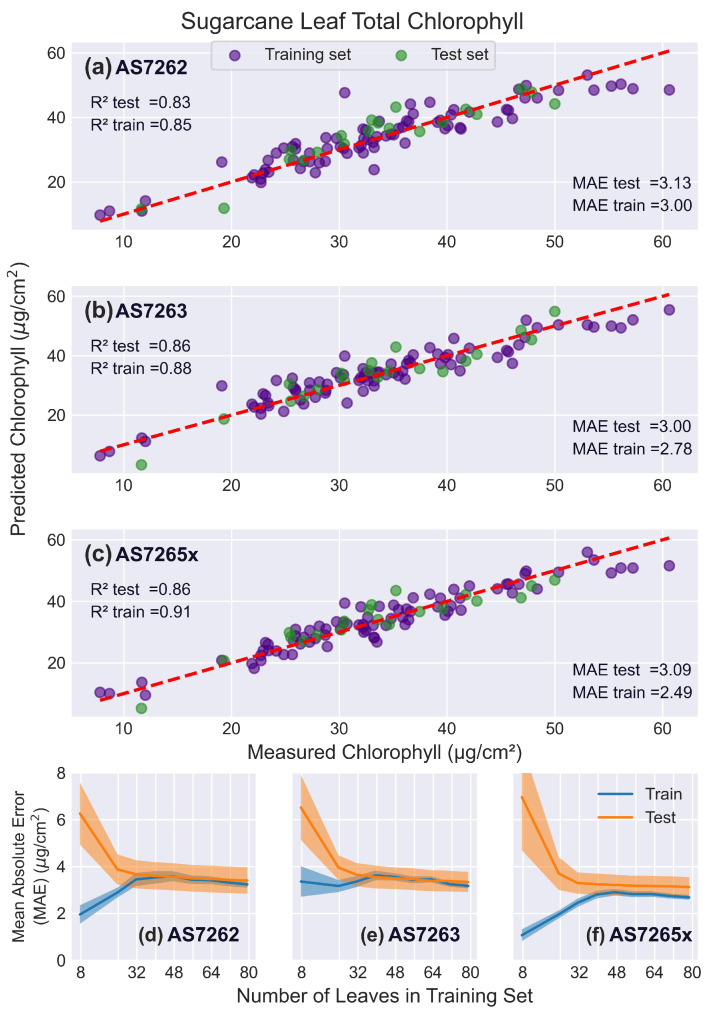
Sugarcane leaf results. PLS model fits to sugarcane leaves. (**a**–**c**) Predicted versus actual chlorophyll levels for the sensors: (**a**) AS7262, (**b**) AS7263, and (**c**) AS7265x, with the red dashed line representing a perfect fit. (**d**–**f**) Learning curves showing the MAE as a function of the number of leaves in the training set, for the sensors: (**d**) AS7262, (**e**) AS7263, and (**f**) AS7265x. Training scores are shown in blue, and test scores in orange, with shaded areas representing the standard deviation.

**Figure 12 sensors-25-02198-f012:**
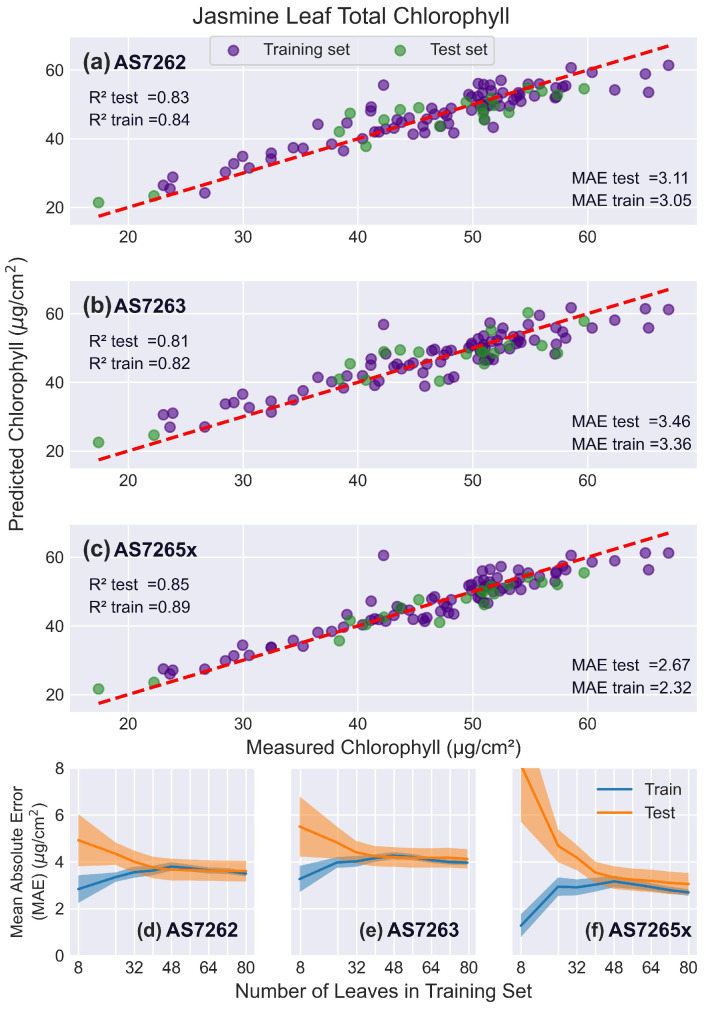
Jasmine leaf results. PLS model fits to jasmine leaves. (**a**–**c**) Predicted versus actual chlorophyll levels for the sensors: (**a**) AS7262, (**b**) AS7263, and (**c**) AS7265x, with the red dashed line representing a perfect fit. (**d**–**f**) Learning curves showing the MAE as a function of the number of leaves in the training set, for the sensors: (**d**) AS7262, (**e**) AS7263, and (**f**) AS7265x. Training scores are shown in blue, and test scores in orange, with shaded areas representing the standard deviation.

**Figure 13 sensors-25-02198-f013:**
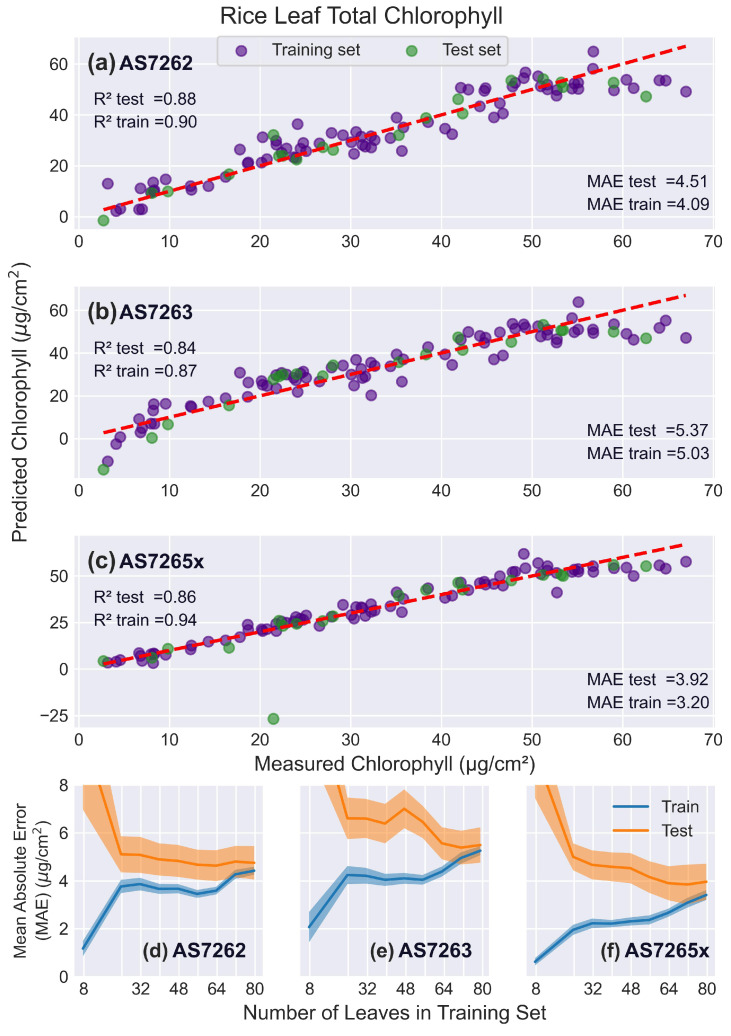
Rice leaf results. PLS model fits to rice leaves. (**a**–**c**) Predicted versus actual chlorophyll levels for the sensors: (**a**) AS7262, (**b**) AS7263, and (**c**) AS7265x, with the red dashed line representing a perfect fit. (**d**–**f**) Learning curves showing the MAE as a function of the number of leaves in the training set, for the sensors: (**d**) AS7262, (**e**) AS7263, and (**f**) AS7265x. Training scores are shown in blue, and test scores in orange, with shaded areas representing the standard deviation.

**Figure 14 sensors-25-02198-f014:**
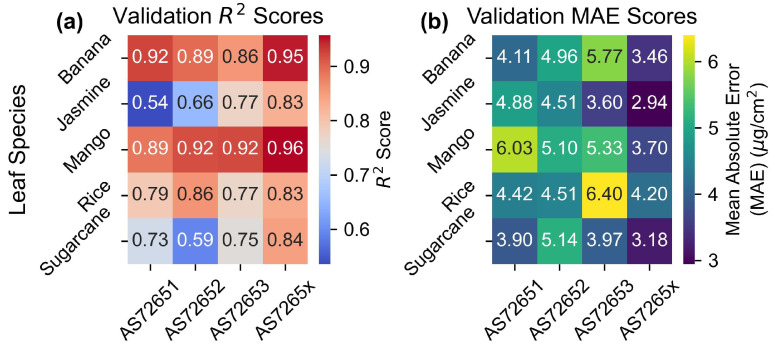
Individual AS7265x chips. Comparison of the performance of the individual chips in the AS7265x sensor with the combined output for (**a**) $R^{2}$ and (**b**) MAE scores. Rice leaves perform better on some individual chips, suggesting performance issues when measuring non-uniform surfaces.

**Table 1 sensors-25-02198-t001:** Optimal number of latent variables for each sensor–leaf combination.

	AS7262	AS7263	AS7265x
Banana	6	5	8
Jasmine	4	5	14
Mango	5	5	11
Rice	6	5	6
Sugarcane	6	5	6

## Data Availability

Raw data, scripts to generate the data and figures used in the manuscript, programs to run the sensors, and GUI used to collect the data are available at https://github.com/KyleLopin/asm_chloro_test (accessed on 25 March 2025).

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
