# Peer review of "Evaluation of Low-Cost Multi-Spectral Sensors for Measuring Chlorophyll Levels Across Diverse Leaf Types"

_sensors, 2025, doi:10.3390/s25072198_

Round 1
Reviewer 1 Report
Comments and Suggestions for Authors
Evaluation of Low-Cost Multi-Spectral Sensors for Measuring Chlorophyll Levels Across Diverse Leaf Type
This study by Lopin et al., addresses an important agricultural challenge of accurate, low cost chlorophyll measurement. It provides valuable insights for small-scale farmers and precision agriculture applications. I’m impressed with the use of Partial Least Squares (PLS) regression and nested cross-validation which strengthens the statistical robustness. Their multiple sensor comparisons viz. AS7262, AS7263, AS7265x provide a well-rounded evaluation. Although the paper is methodologically strong, highly relevant, and well-structured but would benefit from clearer explanations, practical comparisons, and deeper discussion on sensor limitations.
For starters, some sections are overly technical without introductory transitions. I’d add a short explanations at the beginning of each methods section to outline why each step is performed.
The role of sensor geometry and leaf morphology in reducing accuracy is mentioned, but solutions are not explored in depth.
The study does not explicitly compare the AMS sensors to commercial chlorophyll meters like the SPAD or atLEAF. I’d include a brief discussion of how AMS sensors compare in cost, accuracy, and usability against existing commercial devices.
Some figures are dense with information and difficult to interpret at a glance, maybe use simpler labels, bold important trends, or split large tables into smaller, digestible sections.
Please mention full forms as it is difficult to follow. For example, Line 43, what is SPAD meter?
Pay special attention to grammar, composition and avoid awkward phrasing. For example, what is ‘Small scall farmers’, check line 71?
Keep up the good work and best of luck,
Cheers!
Author Response
Comments 1: For starters, some sections are overly technical without introductory transitions. I’d add a short explanations at the beginning of each methods section to outline why each step is performed.
Response 1: We thank the reviewer for their helpful review. In response to the comments, we have added an introductory explanation at the beginning of each method subsection.
Comments 2: The role of sensor geometry and leaf morphology in reducing accuracy is mentioned, but solutions are not explored in depth.
Response 2: We have included a new section at the end of the Discussion section, “Overall Performance and Limitations”, which addresses how sensor geometry and leaf morphology affect sensor performance. This section also suggests the possible solution of adding an optical diffuser, though more research is needed.
Comments 3: The study does not explicitly compare the AMS sensors to commercial chlorophyll meters like the SPAD or atLEAF. I’d include a brief discussion of how AMS sensors compare in cost, accuracy, and usability against existing commercial devices.
Response 3: The “Overall Performance and Limitations” also compares AMS sensors with SPAD and atLEAF sensors.
Comments 4: Some figures are dense with information and difficult to interpret at a glance, maybe use simpler labels, bold important trends, or split large tables into smaller, digestible sections.
Response 4: We are also concerned by the format of our figures. We have tried different formats to optimize clarity while including important information. The current configuration is the best balance we have found among various configurations we have attempted.
Comments 5: Please mention full forms as it is difficult to follow. For example, Line 43, what is SPAD meter?
Pay special attention to grammar, composition and avoid awkward phrasing. For example, what is ‘Small scall farmers’, check line 71?
Response 5: We have expanded all abbreviations upon first, including defining 'SPAD meter' in Line 43. Additionally, we have corrected the grammatical errors pointed out and double-checked the rest of the document for additional improvements.
Reviewer 2 Report
Comments and Suggestions for Authors
The authors test three low-cost devices for performance in measurement of chlorophyll level in leaves of five species, which differ in type, morphology and chlorophyll content. Such measurements are basic for filed crops managements for diagnostic purposes, namely for estimation of nitrogen deficit in plants or overdose of fertilizers'. Studies with use of this type light sensors has been published previously e.g. in "Sensors", that is cited in the manuscript.
The paper presents, however, very reliable statistical approach including Partial Least Squares (PLS) regression model with double-nested cross-validation for model fitting. Moreover, the evaluation of the technique is based on wet destructive analytical method for chlorophyll content estimation, well accepted in plant physiology. This is thus another proof for the usefulness of the approach.
Manuscript is prepared very professionally, starting from technical details description, outlining principles of the device operation, graphical presentation and the method performance validation. Statistics is correctly applied, figures are high quality and well serve illustrating the applications and procedure of the device validation.
Text layout is preserved in accordance with the requirements of the editorial. The language and structure of the article are clear and easily understandable. The relevance of the method and principle of device operation are properly explained.
This paper fits well to the Sensors" scope. The manuscript seems to be ready for publication from the point of methodical correctness.
However, it is not easy to read the manuscript while being overwhelmed by the technical details.
I think that the main manuscript would not lose its value if it contained only the essential content, emphasizing the idea and evidence, while the technical details were partially transferred to a Supplementary file. But this suggestion I leave for the Editor decision and journal preferences.
- What is the main question addressed by the research?
The aim of the research is to compare three commercially available light sensors for measurement of chlorophyll content in leaves. The performance of the sensors is tested on the base of five different types of leaves using appropriate statistical model (Partial Least Squares (PLS) regression model with double-nested cross-validation for model fitting). The validation of the method is referred to photometric analysis of chlorophyll content accepted in plant physiology.
On the base of the analysis the authors recommend all the sensors as acceptable for estimation of nitrogen status of leaves and indicate difference between them.
- Do you consider the topic original or relevant to the field? Does it address a specific gap in the field? Please also explain why this is/ is not the case.
These sensors were tested for their suitability in chlorophyll measurements before and the authors cited appropriated papers.
The comparisons of sensors is made here using different methodological approach, comparing to earlier reports. That is the only novelty of the researches. The statistical analysis is carried out with the due rigor required (removal outliers, model optimalisation and validation)
However, I would treat this work as a confirmatory studies rather than shedding a different light on the problem or filling the gaps in understanding the technology.
- What does it add to the subject area compared with other published material?
Unfortunaley, apart from using a new (in reference to these sensors) methodology (otherwise well known, in general)) I can not find other aspects of novelty.
- What specific improvements should the authors consider regarding the methodology?
I have no critical comments about the methodology
- Are the conclusions consistent with the evidence and arguments presented and do they address the main question posed? Please also explain why this is/is not the case.
The conclusion is consistent with addressed question and performed analysis. The authors provide precise evaluation of the sensors for suggested applicability.
- Are the references appropriate?
Yes, the authors cited papers which referred to the simitar problem
- Any additional comments on the tables and figures
All tables and figures were prepared carefully and in my opinion correctly.
General comment:
This paper is well prepared with all required scientific rigors, and well written, in my opinion ready for publications.
However, I should emphasise that novelty is average (a confirmatory studies). The novely my be attributed only to different methodology used to compare and validate the sensors.
Referring to the recommendation for publication:
I would consider the acceptance dependent on the threshold of requirements established by the Journal. This is well prepared confirmatory paper of limited scientific novelty.
Author Response
Response to Reviewer 2
Comments 1: This paper fits well to the Sensors" scope. The manuscript seems to be ready for publication from the point of methodical correctness.
However, it is not easy to read the manuscript while being overwhelmed by the technical details.
I think that the main manuscript would not lose its value if it contained only the essential content, emphasizing the idea and evidence, while the technical details were partially transferred to a Supplementary file.
Response 1: We appreciate the reviewer’s comments. In response, we have moved the sections on electrical measurements, chlorophyll reference measurements, and the statistical analysis of the best conditions to Supplemental Information. This revision makes the manuscript more concise while preserving the detailed information for interested parties.
Reviewer 3 Report
Comments and Suggestions for Authors
Title: Evaluation of Low-Cost Multi-Spectral Sensors for Measuring Chlorophyll Levels Across Diverse Leaf Types
General
The paper, as its names suggest presents a study of a few optical sensors for the evaluation of chlorophyl in plants. It is an important parameter as mentioned in numerous papers and reviews and there are even a few commercially available devices that can be purchased “off the shelf”. For example, CCM-200 (Opti-Sciences, Tyngsboro, Massachusetts, USA), and the SPAD- 502 (Minolta Camera Co., Osaka, Japan). However, chlorophyll levels are not affected only by nutrients as they also depend on the plant’s tress due to pollution for example. Moreover, the relation between the relation between chlorophyll meter readings and leaf chlorophyll concentration, nitrogen status, and crop yield is still under study. Nevertheless, chlorophyll metering is still a necessary and useful measurement. Fluorescence (i.e. Chlorophyll fluorescence (ChlF) is still one of the most useful methods to assess the monitor chlorophyll in plants. As written in the literature “the analysis of fluorescence signals provides detailed information on the status and function of Photosystem II (PSII) reaction centers, light-harvesting antenna complexes, and both the donor and acceptor sides of PSII”
Hence the question is what is the scientific value of publishing a study of a known method using the AS7262, AS7263, and AS7265x spectrometers? And available in the market.
I think there is value, although not so much highlighted by the authors. Note that recently a few other companies started to deliver low-cost, MEMS-based multi and hyper-spectral spectrometers for the visible and NIR. Such devices are usually very small, inherently low-cost, and with the potential to become even cheaper. Note that low cost is not enough since it should be low power and interface to the network. Such miniature devices open the door to a much wider use of field deployable sensors providing significantly more data. Combining it with the new trend to use field deployable sensors in a network connected to the cloud (e.g. Internet of Things) and the increasing use of AI/ML algorithms making data-driven agriculture more effective there is value in publishing such a paper.
The authors also stress the issue of “low cost “intensively. The value of the paper must be in its engineering value as it is very thorough, showing in detail the measurements and analysis methods, and can be used for other researchers developing similar systems. Such systems can be much cheaper in the future.
Specific
- The paper should discuss also the usability of such systems in the field: not only cost but also power, ease of use, and possible automation and networking. A lot of the work in the paper requires manual work in sample preparation and analysis. They ignore that cost. Can they automate everything?
- The paper covers a lot of ground and there is a lot of redundancy in the measurements. They present data for numerous plants. To make their point they should consider shortening their work significantly. It is an engineering q technical paper so there is no need for repeated experiments unless there are some fundamental differences.
References
- Murchie, E.H. and Lawson, T., 2013. Chlorophyll fluorescence analysis: a guide to good practice and understanding some new applications. Journal of experimental botany, 64(13), pp.3983-3998.
- Oxborough, K., 2004. Imaging of chlorophyll a fluorescence: theoretical and practical aspects of an emerging technique for the monitoring of photosynthetic performance. Journal of Experimental Botany, 55(400), pp.1195-1205.
- Andrianto, H. and Faizal, A., 2017, October. Measurement of chlorophyll content to determine nutrition deficiency in plants: A systematic literature review. In 2017 International Conference on Information Technology Systems and Innovation (ICITSI)(pp. 392-397). IEEE.
- Talebzadeh, F. and Valeo, C., 2022, April. Evaluating the effects of environmental stress on leaf chlorophyll content as an index for tree health. In IOP Conference Series: Earth and Environmental Science(Vol. 1006, No. 1, p. 012007). IOP Publishing.
- Talebzadeh, F. and Valeo, C., 2022, April. Evaluating the effects of environmental stress on leaf chlorophyll content as an index for tree health. In IOP Conference Series: Earth and Environmental Science(Vol. 1006, No. 1, p. 012007). IOP Publishing.
- Kalaji, H.M., Jajoo, A., Oukarroum, A., Brestic, M., Zivcak, M., Samborska, I.A., Cetner, M.D., Łukasik, I., Goltsev, V. and Ladle, R.J., 2016. Chlorophyll fluorescence as a tool to monitor physiological status of plants under abiotic stress conditions. Acta physiologiae plantarum, 38, pp.1-1
- Henriques, F.S., 2009. Leaf chlorophyll fluorescence: background and fundamentals for plant biologists. The Botanical Review, 75, pp.249-270
- Ata-Ul-Karim, S.T., Cao, Q., Zhu, Y., Tang, L., Rehmani, M.I.A. and Cao, W., 2016. Non-destructive assessment of plant nitrogen parameters using leaf chlorophyll measurements in rice. Frontiers in Plant Science, 7, p.1829.
Author Response
Response to Reviewer 3
Comments 1: The paper should discuss also the usability of such systems in the field: not only cost but also power, ease of use, and possible automation and networking. A lot of the work in the paper requires manual work in sample preparation and analysis. They ignore that cost. Can they automate everything?
Response 1: We appreciate the reviewer’s constructive comments. In response, we have added a new section at the end of the Discussion section, “Overall Performance and Limitations,” to discuss the sensors’ current usability. This section discusses the manual work required to make the calibration models but highlights that the sensors can quickly and easily be used for many readings once calibrated.
Comments 2: Hence the question is what is the scientific value of publishing a study of a known method using the AS7262, AS7263, and AS7265x spectrometers? And available in the market.
I think there is value, although not so much highlighted by the authors. Note that recently a few other companies started to deliver low-cost, MEMS-based multi and hyper-spectral spectrometers for the visible and NIR. Such devices are usually very small, inherently low-cost, and with the potential to become even cheaper. Note that low cost is not enough since it should be low power and interface to the network. Such miniature devices open the door to a much wider use of field deployable sensors providing significantly more data. Combining it with the new trend to use field deployable sensors in a network connected to the cloud (e.g. Internet of Things) and the increasing use of AI/ML algorithms making data-driven agriculture more effective there is value in publishing such a paper.
Response 2: Additionally, we have mentioned in the “Overall Performance and Limitations” subsection how the sensor design allows for the potential incorporation of different connectivity options or AI integration in future devices, though further research is needed. We are currently working on developing a device to improve sensor usability. Also, we are comparing the AMS sensors to a MEMS-based spectrometer in another application, where the AMS sensors appear less sensitive to mechanical disturbances.
Comments 3: The paper covers a lot of ground and there is a lot of redundancy in the measurements. They present data for numerous plants. To make their point they should consider shortening their work significantly. It is an engineering q technical paper so there is no need for repeated experiments unless there are some fundamental differences.
Response 3: We acknowledge that certain redundancies in our work have limited value from an engineering perspective; however, measuring the range of errors across different systems can provide useful insights. However, we also aimed to create a technical guide for agriculturalists and plant biologists on sensors performance across different leaf types. To enhance accessibility, we have structured the manuscript to group engineering and agricultural parts together, allowing readers to focus on the aspects most relevant to their interests.
Round 2
Reviewer 3 Report
Comments and Suggestions for Authors
The paper still has a few fundamental problems.
A. The spectrum figures and the X-Axis labels are unclear. For example:
Figure 4: The X axis is 500, 550, 570, 600, and 650 nm,,,, why add the 570? Over the entire paper, they are not specific about why they choose the wavelength the way they choose. Was it due to? The availability of the spectral band? Was there a choice? How does this affect
They also present to the right a color schema demonstrating the concentration of the chlorophyll in micrograms/cm square. How do they calculate it, and it is unclear why this is a function of the spectrum?
Figure 6: The X-Axis has 12 numbers while the plot has 17 regions (laterally).
B. I still advise them to shorten the paper significantly. There is still a lot of redundant information that has little relevance to the main subject, for example, Table 1; is it necessary? I do not see a fundamental difference between the various examples they present.
A lot of material can be removed to supplementary files or just removed.
Author Response
Comments 1: A. The spectrum figures and the X-Axis labels are unclear. For example:
Figure 4: The X axis is 500, 550, 570, 600, and 650 nm,,,, why add the 570? Over the entire paper, they are not specific about why they choose the wavelength the way they choose. Was it due to? The availability of the spectral band? Was there a choice? How does this affect
Response 1: Thank you for your observation. Yes, the x-axis represents the spectral channels, we have clarified this in lines 156-162 of the Methods section and in the legends for Figure 4, 5, and 6. The 570 nm band is based on the spectral bands of the sensor, which are not user-selectable..
Comments 2: They also present to the right a color schema demonstrating the concentration of the chlorophyll in micrograms/cm square. How do they calculate it, and it is unclear why this is a function of the spectrum?
Response 2: Chlorophyll levels were measured following the procedure described in the Methods section, "Leaf Chlorophyll Measurements". The raw spectral and reflectance data are displayed with chlorophyll levels because it provides an intuitive visual check to see if there are visual differences in the raw data before it is processed. By visualizing chlorophyll levels and spectral reflectance in a single graph, we can see how the spectrum changes in response to chlorophyll levels. If the different chlorophyll levels can visually be seen to have quantitative differences in the spectrum, it's evidence that a model should be able to effectively fit the relationship. Also, the leaf spectra have to be plotted in some color, so why not use different shades of green that reflect their actual color?
Comments 3: Figure 6: The X-Axis has 12 numbers while the plot has 17 regions (laterally).
Response 3: Figure 6 contains 18 grid lines for the 18 spectral channels of the AS7265x (this is now mentioned in the Figure 6 legend). However, not all spectral channel labels are displayed due to space constraints, as they would start to overlap.
Comments 4: B. I still advise them to shorten the paper significantly. There is still a lot of redundant information that has little relevance to the main subject, for example, Table 1; is it necessary? I do not see a fundamental difference between the various examples they present.
A lot of material can be removed to supplementary files or just removed.
Response 4: We have removed Table 1.
While we acknowledge some redundancy exists, we feel it adds value by including detailed discussions on model selection, outlier removal and error quantification in reference measurements. Standard practice often involves averaging spectra and reference measurements. This approach merely reduces errors but does not eliminate them in the way a robust outlier detection method does.
Additionally, including a diverse set of leaves reveals important nuances in the data. For example, comparing mango and jasmine leaves highlights a better R2 fit for mango, while jasmine has a lower mean absolute error, despite jasmine having more error in the reference chlorophyll levels. This adds redundancy in having multiple metrics and multiple leaves, but we feel this highlights an important distinction: while the field prioritizes R2 scores as a measure of sensor performance, the objective should be to minimize measurement errors in the parameter of interest.
We have also labeled the subsections clearly, enabling readers to skip parts that are not relevant to them